# Brief communication: An empirical relation between center frequency and measured thickness for radar sounding of temperate glaciers

Joseph A. MacGregor[1], Michael Studinger[1], Emily Arnold[2,3], Carlton J. Leuschen[3], Fernando Rodríguez-Morales[3], John D. Paden[3]

[1]Cryospheric Sciences Laboratory (Code 615), NASA Goddard Space Flight Center, Greenbelt, Maryland, 20771, United States of America

[2]Aerospace Engineering Dept., The University of Kansas, Lawrence, Kansas, 66045, United States of America

[3]Center for Remote Sensing of Ice Sheets, The University of Kansas, Lawrence, Kansas, 66045, United States of America

*Correspondence to*: Joseph A. MacGregor (joseph.a.macgregor@nasa.gov)

**Abstract.** Radar sounding of the thickness of temperate glaciers is more challenging than for polar ice sheets, due to the former's greater volume scattering (englacial water), surface scattering (crevasses and debris) and dielectric attenuation rate (warmer ice). Lower frequency (~1–100 MHz) radar sounders are commonly deployed to mitigate these effects, but the lack of a synthesis of existing radar-sounding surveys of temperate glaciers limits progress in system and survey design. Here we use a recent global synthesis of glacier thickness measurements to evaluate the relation between the radar center frequency and maximum thickness. From a maximum reported thickness of ~1500 m near 1 MHz, the maximum thickness sounded decreases with increasing frequency by ~500 m per frequency decade. Between 25–100 MHz, newer airborne radar sounders generally outperform older, ground-based ones, so radar-sounder success is also influenced by system design and processing methods. Based on globally modeled glacier thicknesses, we conclude that a multi-element airborne radar sounder with a center frequency of ≤ 30 MHz could survey most temperate glaciers more efficiently than presently available systems.

## 1    Introduction

Measuring the thickness of Earth's mountain glaciers is essential for advancing understanding of their volume, flow and future amid ongoing anthropogenic warming, consequent mass loss and contribution to sea-level rise (e.g., Farinotti et al., 2019; Zemp et al., 2019). Radar sounding is unambiguously the preferred method for most surveys of glacier thickness, due to its logistical efficiency relative to other methods while achieving satisfactory precision and accuracy (Welty et al., 2020). However, most mountain glaciers outside the polar regions are either observed or assumed to be temperate (i.e., at or near the pressure-melting point throughout), and radar sounding of such ice is substantially more challenging than for polar ice sheets (where most of ice column well below the pressure-melting point) or polythermal glaciers (e.g., Pritchard et al., 2020). Three

main factors conspire to cause this challenge: 1. Englacial water in pore spaces or fractures, increasing volume scattering (e.g., Fountain et al., 2005); 2. More common crevassing and supraglacial debris, increasing surface scattering (e.g., Herreid and Pellicciotti, 2020); and 3. Warmer ice, increasing the englacial dielectric attenuation rate (e.g., Stillman et al., 2013).

Watts and England (1976) described what may be the primary challenge in radar sounding of temperate ice: meter-scale, water-filled englacial cavities that efficiently scatter incident radio waves where the ratio of those cavities' radius to the radar's
englacial wavelength exceeds ~0.1. Their analysis favored center frequencies ≤ ~10 MHz to increase the signal-to-clutter ratio between the ice–bed reflection (signal) and any cavity-induced volume scattering (clutter). Their lucid description of this challenge motivated the development of numerous "low-frequency" radar sounders (e.g., Watts and Wright, 1981; Fountain and Jacobel, 1997; Conway et al., 2009; Mingo and Flowers, 2010; Rignot et al., 2013; Arnold et al., 2018; Björnsson and Pálsson, 2020). However, subsequent advances in available hardware, system design and processing demonstrated that higher-
frequency (> 10 MHz) radar sounders can also sound hundreds of meters of temperate ice (e.g., Rutishauser et al., 2016; Langhammer et al., 2019; Pritchard et al., 2020). No synthesis yet exists of the success of these radar sounders as a function of center frequency, limiting our ability to identify outstanding opportunities in system and survey design for more efficient sounding of temperate glaciers. Here we evaluate past and potential radar-sounder performance by examining recent global syntheses of both observed and modeled glacier thickness.

**2      Data and methods**

We primarily use three data sources in this study: 1. The maximum reported ice thickness for individual surveys compiled in the Glacier Thickness Database (GlaThiDa) version 3.1.0 (Welty et al., 2020); 2. The consensus modeled thickness estimates for all glaciers on Earth (Farinotti et al., 2019); 3. The regions, glacier locations, areas and identification numbers of the Randolph Glacier Inventory (RGI) version 6 (RGI Consortium, 2017).
We extend GlaThiDa with one additional field: the center frequency of the deployed radar sounder for surveys that used this method (Supplementary Information). While englacial wavelength is likely the more fundamental physical property affecting radar-sounding success (Watts and England, 1976), it is the center frequency that is most often reported and straightforward to synthesize. In most cases, we could determine the value of this field directly from other metadata provided by GlaThiDa. In dozens of cases, we reviewed the primary source for the survey cited by GlaThiDa to determine the center
frequency, assuming that the radar sounder with the lowest center frequency reported was that which detected the maximum ice thickness reported by GlaThiDa, if that was not stated explicitly. In 210 cases (4% of radar surveys in GlaThiDa), we were unable to determine the center frequency of the deployed radar sounder.

For ground-based surveys, a potential source of ambiguity is whether the reported center frequency of the transmitted radio wave is for air or ice, which can be challenging to determine from available metadata. This ambiguity arises because the
wavelengths generated by a given dipole antenna are the same if that antenna is on the ground or airborne, but their frequencies will differ depending on their velocity within the surrounding media. If the reported center frequency is for ice, then it will be

artificially lower by a factor of ~1.79 (the radio-frequency index of refraction for meteoric ice) than that of most other reported frequencies (e.g., Mingo and Flowers, 2010). For simplicity, here we assume that the reported center frequency of all ground-based surveys is that through air, so that they can be compared directly to airborne surveys.

For all GlaThiDa entries that reported a maximum thickness for a presumed temperate glacier at the upper end of the range reported for that frequency, we reviewed the original study to validate the value reported in GlaThiDa. For all glaciers with reported raw glacier-thickness data in GlaThiDa but no reported maximum value, which are mostly attributed to Rignot et al. (2013) and Rutishauser et al. (2016), we calculate the maximum thickness directly from those raw data. Finally, we adjust GlaThiDa's survey method field to further distinguish airborne radar-sounding surveys between helicopter and fixed-wing

surveys.

We distinguish between regions that are most likely to contain temperate glaciers versus those that mostly contain polythermal or polar glaciers, while recognizing that substantial uncertainty remains in the thermal structure of many mountain glaciers (e.g., Wilson and Flowers, 2013). We assume that temperate glaciers are predominant within all RGI regions except Arctic Canada (03 and 04), Greenland (05), Svalbard (07), the Russian Arctic (09), and the Antarctic and sub-Antarctic (19).

Glaciers situated at $\geq 67°$ latitude (e.g., McCall Glacier, Alaska, and Storglaciären, Sweden), along with known polythermal glaciers in presumably otherwise temperate regions, are excluded from the remaining set: 1. Hazard, Rusty and Trapridge glaciers, Canada (Narod and Clarke, 1980); 2. Gornergletscher, Switzerland (Rutishauser et al., 2016); and 3. Khukh Nuru Uul, Mongolia (Herren et al., 2013). In addition to the resulting 324 frequency–thickness pairs for glaciers in temperate regions from GlaThiDa, we also show values from ground-based surveys recently completed for the Himalaya (440 m at 3.5 MHz; Pritchard et al., 2020), western Canada (318 m at ~18 MHz in air; Pelto et al., 2020) and the Bagley Icefield, Alaska (1460 m

at 1.875 MHz; M. Truffer and J. W. Holt, pers. comm., 2020).

## 3 Results

Fig. 1a shows the relation between reported maximum ice thickness and the center frequency in air of the deployed radar sounder, further differentiated by survey year (if known or reported) and platform type (for presumed temperate glaciers in

GlaThiDa: 155 ground-based, 102 helicopter and 67 fixed-wing). A general trend of decreasing measured thickness with increasing frequency is apparent. While qualitatively recognized by many practitioners and sometimes investigated directly for the purposes of system and survey design (e.g., Watts and Wright, 1981; Rutishauser et al., 2016; Pritchard et al., 2020), this relation has not been quantified previously across the full spectrum of frequencies used to sound terrestrial ice masses (~1–1000 MHz).


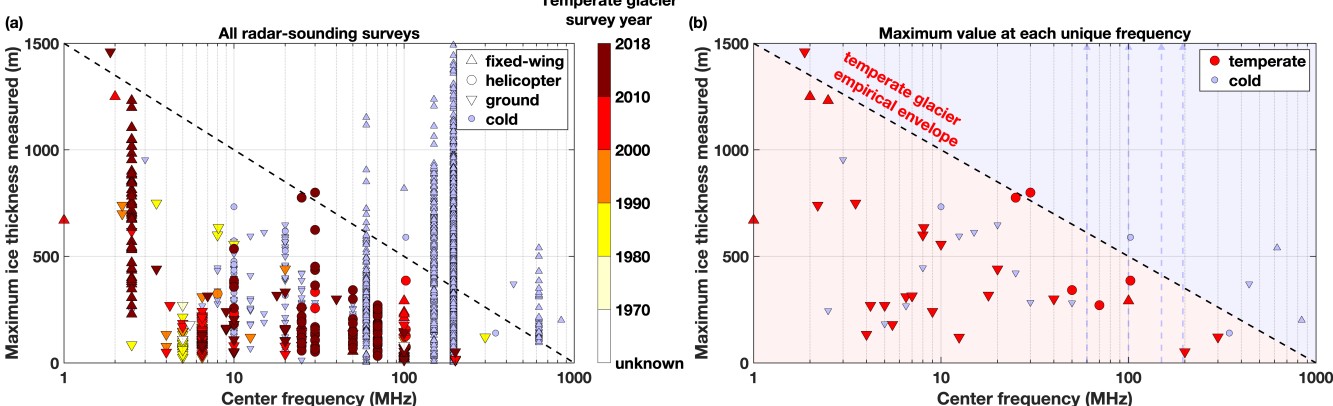

**Figure 1:** (a) Synthesis of all GlaThiDa v3.1.0 reported maximum ice thicknesses versus the center frequency of the deployed radar sounder; and (b) a subset of the same dataset showing only the maximum value for each unique frequency (temperate or "cold"). "Cold" glaciers are either those from the polar regions or those in presumed temperate regions that are known to be polythermal. Symbol shape indicates platform type. Black dashed line represents the empirical envelope of apparent best sounding performance for temperate glaciers. In panel (a), for presumed temperate glaciers, color indicates survey period (if reported). In panel (b), the maximum measured ice thickness for "cold" glaciers at four frequencies (60, 100, 150 and 195 MHz) exceeds the maximum value on the vertical axis (1500 m), represented by vertical blue arrows.

Most radar-sounding surveys of temperate glaciers before 2000 (55 of 63) deployed systems with frequencies below 20 MHz, while about half (138 of 258) of modern surveys (2000–onward) used higher frequencies (≥ 20 MHz). At very low frequencies (< 3 MHz) and at higher ones (> 20 MHz), it is these modern radar sounders that sound unusually large ice thicknesses compared to older surveys at nearby frequencies. However, between 3–20 MHz ground-based surveys are predominant and sound the thickest ice. Between 25–100 MHz, newer airborne systems tend to outperform ground-based systems at comparable frequencies, and helicopter-borne systems tend to outperform fixed-wing systems (e.g., 100 MHz), presumably due to the former's slower platform speed and potentially lower altitude above ground level (Fig. 1b). Higher frequencies (≥ 60 MHz) can sound much thicker polythermal or polar ice than has been achieved for temperate glaciers, but this relative performance advantage for colder ice is reduced significantly below ~30 MHz, which may be due to lower antenna gain and increased environmental interference.

This synthesis contains multiple sampling biases: 1. A radar sounder can only sound temperate ice as thick as the glaciers it surveys, so reported thicknesses are potentially underestimates relative to an individual system's true capability; 2. Most radar-sounding surveys have rarely known beforehand where ice thickness is predicted to be greatest or its expected value there, so for a given glacier this location may not have been surveyed or a suitable radar sounder may not have been selected. (this situation has ameliorated recently with the advent of globally modeled glacier thicknesses, e.g., Farinotti et al., 2019); 3. Ground-based surveys can only occur where it is safe to do so; 4. Some surveys were performed during the summer, when englacial water is likely more abundant and hinders radar performance; 5. Some reported center frequencies for ground-based surveys may be for ice, rather than for air as we have assumed (Sect. 2); and 6. Some glaciers we assume are temperate – due to their regional setting and the lack of contraindicating observations – may not be temperate. The first five biases are likely

negative, i.e., they induce an underestimate of the maximum ice thickness that could be sounded at a given frequency, while the sixth bias is positive with the opposite effect. Separately, off-nadir surface clutter is a well-known source of ambiguity in identification of the ice–bed reflection within valley glaciers (e.g., Holt et al., 2006), and most surveys had no direct method for surface-clutter discrimination (e.g., Conway et al., 2009; Rignot et al., 2013), but its bias can be positive or negative depending on glacier geometry.

Based on this synthesis, we identify a simple envelope of maximum ice thickness sounded for temperate glaciers across more than two decades of frequency range spanned by deployed radar sounders (Fig. 1b). Within this frequency range, the maximum possible ice thickness that can be sounded decreases at a rate of ~500 m per frequency decade, descending from ~1500 m at 1 MHz. For the frequency range shown (1–1000 MHz), this relation is equivalent to $H_{max} = 1500 - 500 \log_{10} f$, where $H_{max}$ is the maximum ice thickness in meters and $f$ is the center frequency in megahertz. This empirical envelope has several limitations: 1. It is simply a linear relation between ice thickness and logarithmic frequency, which was selected because it captures the predominant trend and does not overfit thickness maxima with as-of-yet unjustified complexity; 2. No meaningful uncertainty bounds can be specified presently; and 3. Two surveys report sounding temperate ice slightly thicker than this envelope suggests. The envelope's value lies in its indirect synthesis of the previously mentioned factors that challenge sounding of temperate ice, its identification of frequencies for radar sounders that may be performing near a natural or present technical limit (e.g., ~2–3, ~25–30 and 100 MHz), and others where radar sounders have either underperformed or may be unusually challenged by temperate ice (~5–20 MHz).

Using this empirical envelope, we can then crudely relate the maximum modeled thickness distribution for each predominantly temperate glacierized region to a maximum possible radar-sounder frequency (Fig. 2). We consider only larger glaciers (RGI-reported area $\geq 5$ km$^2$) in each region that are more likely to be targeted for radar-sounding surveys. This comparison highlights the range of frequencies that could possibly sound most glaciers in these regions under ideal conditions, which is larger than assumed previously (Watts and England, 1976). For most temperate regions (10 of 13), this analysis suggests that a modern $\leq 100$-MHz radar sounder could potentially sound the maximum ice thicknesses of 95% of their glaciers ($< 500$ m; Fig. 2). For Alaska, Iceland and the Southern Andes, a lower-frequency ($\leq 30$ MHz) radar sounder remains necessary for many glaciers, as is observed in practice (e.g., Conway et al., 2009; Björnsson and Pálsson, 2020).

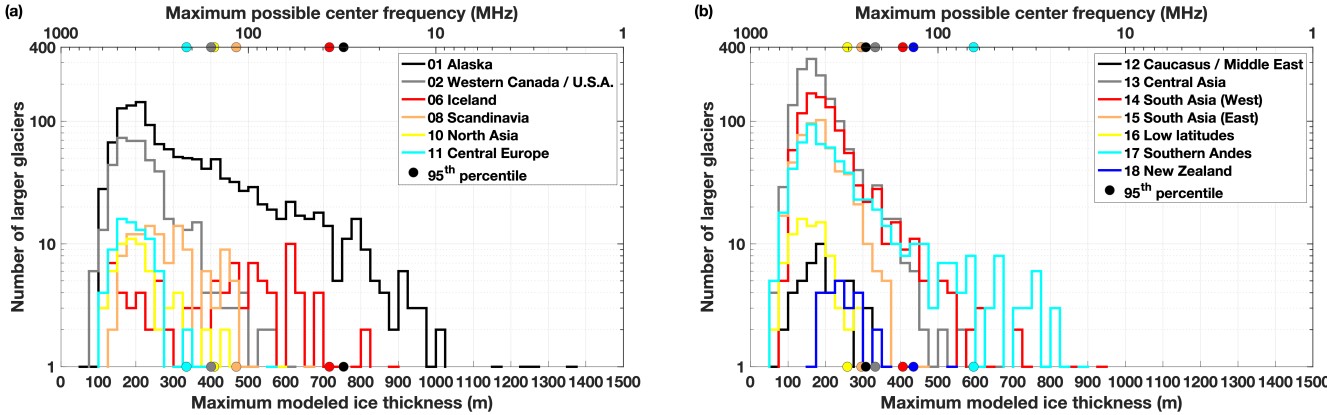

**Figure 2:** Distribution (lines) and 95th percentile (circles) of maximum modeled ice thickness for all larger (≥ 5 km²) glaciers in RGI regions assumed to contain mostly temperate glaciers. Bin interval is 25 m. Upper horizontal axis is equivalent to the empirical envelope in Fig. 1, e.g., for a glacier whose maximum ice thickness is ~500 m, the maximum center frequency of a radar sounder that could possibly sound that glacier under ideal conditions is ~100 MHz.

We note two contrasting caveats to this analysis: 1. While the modeled maximum thicknesses do not appear to be biased significantly relative to measured maxima (+62 ± 197 m for the 36% of GlaThiDa surveys we could confidently match using glacier names to a modeled glacier in the RGI inventory), it is not uncommon for radar-sounding surveys of ice masses to report ice thicknesses greater than predicted beforehand, so any survey design based on Fig. 2 should assume a negative model bias and err on the side of a lower frequency; and 2. The maximum modeled thickness is generally only reached at a single point on a glacier, so surveys aiming to measure glacier volume must also consider the ability to resolve smaller thicknesses at a satisfactory resolution. This trade-off could favor a higher center frequency, for which a larger bandwidth is easier to achieve, potentially resulting in a finer range resolution.

## 4 Discussion and conclusions

Our evaluation of GlaThiDa-compiled thicknesses challenges the conventional wisdom that only low-frequency (≤ 10 MHz) radar sounders are suitable for sounding temperate glaciers that are hundreds of meters thick. An empirical envelope derived from this synthesis suggests that – for most temperate glaciers on Earth – higher frequencies are indeed appropriate for radar sounding thereof, assuming suitable system design. This envelope suggests a higher upper limit (~30 MHz) on potentially suitable radar sounders for sounding temperate ice up to ~700 m thick (Fig. 1), but a more conservative interpretation still suggests that ~500 m is possible at this frequency and could be suitable for sounding the large majority of Earth's glaciers (Fig. 2). Recent advances in hardware and system design could further increase this range, e.g., solid-state transmit/receive switches, higher peak-transmit powers and platform-aware numerical optimization of antenna configurations (Arnold et al., 2020). While a full accounting of the physical underpinnings of the simple empirical envelope we identified

was beyond the scope of this study, the envelope cannot be explained by the relatively weak dispersion of the radio-frequency dielectric attenuation rate (MacGregor et al., 2015), so volume and surface scattering are the more likely controls.

We suggest using this empirical envelope as a guide to help balance a radar-sounding survey's scientific objectives with the system to be deployed. For example, selecting a system toward the low end of radio-frequency range considered (e.g., $\leq 5$ MHz) is only necessary if the objective is to sound the thickest temperate ice in a handful of regions. Alternatively, a higher frequency ($\geq 20$ MHz) radar sounder could be deployed to more finely resolve glacier volume at the possible expense of not sounding the thickest portions of a larger glacier. Not all glaciers are created equal, and in some cases an exceptionally

crevassed or rough surface, thicker supraglacial debris, abundant englacial water or simply exceptionally thick ice will continue to necessitate the use of lower frequencies to ensure successful sounding.

Models of glacier thickness are increasingly incorporating mass conservation and global satellite remote-sensing datasets, but there remains an outstanding need for additional thickness measurements to both validate and refine those models' underlying assumptions (e.g., Farinotti et al., 2019; Pelto et al., 2020). The spatial coverage of even the most extensive ground-

based radar-sounding survey of a glacierized region can be easily dwarfed by a single airborne survey – assuming that survey's radar sounder can match the performance of the ground-based system (e.g., Pritchard et al., 2020). Langhammer et al. (2019) demonstrated recently that two orthogonal pairs of helicopter-towed 25-MHz antennas can better sound temperate glaciers because the pseudo-scalar sum of the antenna response is less sensitive to unfavorable bed geometry. Alternatively, a slightly higher frequency (e.g., 30 MHz) translates to a dipole antenna that is $\leq 5$ m long, a dimension that is sufficiently small that

several such elements could potentially be mounted on a fixed-wing aircraft (e.g., Arnold et al., 2018). More than two antenna elements in a plane perpendicular to the platform's direction of travel are essential to resolving cross-track ambiguity in the direction of arrival of coherently recorded reflections, i.e., "swath mapping" (e.g., Holschuh et al., 2020). Such a capability could be quite useful for sounding mountain glaciers, as it could transform some of what is presently discarded as noise (discontinuous near-bed subsurface reflections not attributable to surface clutter) into useful signal (geolocated, off-nadir ice–

bed reflections) and substantially expand our ability to measure glacier thickness efficiently.

**Author contribution.** JAM initiated this study, led the analysis and drafted the manuscript. MS aided in the study design, analysis and manuscript preparation. EA, CJL, FRM and JDP aided in the analysis and manuscript preparation.

**Data and code availability.** The Supplementary Information associated with this article contains the new metadata generated for this study, which is in a format similar to GlaThiDa v3.1.0. The analysis was performed using MATLAB R2021a and both

its Mapping and Statistics & Machine Learning toolboxes. The script used to perform the analysis and generate the figures is available at: https://github.com/joemacgregor/misc/blob/main/temperate_sounding.m.

**Competing interests.** JAM is a member of the editorial board of this journal.

**Acknowledgements.** We thank the NASA/GSFC Strategic Science program for supporting this study. We thank the authors of and contributors to GlaThiDa, Farinotti et al. (2019) and the RGI Consortium for making our study possible through open
distribution of well-documented, homogenized datasets. We thank R. Hale, J.W. Holt, H. Jiskoot, H. Machguth, L. Mingo and M. Truffer for valuable discussions. We thank scientific editor D. Farinotti, M. Grab and an anonymous reviewer for constructive reviews that improved this manuscript.

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
