# Peer review of "Brief communication: An empirical relation between center frequency and measured thickness for radar sounding of temperate glaciers"

_The Cryosphere, 2021_

## Referee Comment (RC1)

**Comment on "Brief communication: An empirical relation between center frequency and measured thickness for radar sounding of temperate glaciers" by Joseph A. MacGregor et al.**

Review by Melchior Grab
February 18, 2021

The relation between the central frequency of impulse radars and the penetration depth of the radar waves into glacier ice is complex, because it depends on various factors given by the nature of the glaciers such as water content in the ice, roughness of the glacier bed, or size and spatial density of crevasses, and on instrument parameters like the height of instrument employment above the ice or the transmitter power. Due to this complexity, no simple formula is known up to date for evaluating the central frequency required to investigate a glacier with a certain ice thickness.

In recent years, a data base has been compiled by Welty et al. (2020), containing an unprecedented number of ice thickness data from all over the world obtained from radar campaigns. In their study, MacGregor et al. made use of these data and established a simple log-linear relationship, that indicates the maximum ice thicknesses that can be surveyed with a given frequency. According to the authors and also to my knowledge, this is the first time, such an investigation has been done based on such a comprehensive amount of data.

**General Comment:**
The study is well written and concise, therefore well-suited for a 'brief communication' in The Cryosphere journal. I agree with the authors (line 109) that a more complex relationship is not justified under consideration of the potential biases listed on lines 93-103.

I recommend the authors add a few sentences at the end of the article, where they summarize how they interpret their empirical relationship between sounding depth and frequency and how they would like to see it be used in practice (See also my minor comment about lines 118-120.

**Minor Comments:**
**Line 18:** *"Newer airborne radar sounders generally outperform older, ground-based ones at comparable frequencies"*:
This finding cannot be deduced from the results presented in this manuscript. See comments about lines 87-88 below.

**Line 82:** Caption of Figure 1-a: Only after studying Figure 1-b it becomes clear that grey symbols are data from cold glaciers. I suggest to add a comment in the caption.

**Lines 87-88:** "*Especially at higher frequencies (≥ 20 MHz), newer radar sounders (2000–onward) outperform older ones, which favored lower frequencies (≤ 10 MHz).*":

This cannot be deduced from the data shown in Figure 1a. There are only two data points acquired before 2000 at frequencies ≥ 20 MHz: the two with GlaThiDa id's 334 and 2100 (see Fig A). The data point with id 334 is showing an ice thickness which is larger than all the thicknesses from newer campaigns at the same frequency and for temperate ice. Also in comparison with newer measurements at other frequencies the

measured ice thickness of point 334 seems to be larger than average. The other data point with id 2100 shows a lower thickness of only around 120 meters but is even closer to the envelope. Either not all the data is shown (are there some symbols hidden by others in Fig 1a?) that leads to the conclusion that newer campaigns outperform older ones or this conclusion is wrong.

**Lines 94-96:** "*2. While the situation is changing with the advent of globally modeled glacier thicknesses (e.g., Farinotti et al., 2019), radar-sounding surveys have historically not always known beforehand where ice thickness is predicted to be greatest, nor its expected value*"
The reader can only guess how this leads to the "likely negative bias" to which you refer to on line 99. Please provide further explanations.
Do you mean that ice thickness models provide information for planning radar surveys? If the glacier exhibits larger ice thicknesses than expected and researchers accidently use a radar system with too low penetration depth, their instruments perform at the limit and we would not expect a negative bias. Alternatively, researchers might acquire data while choosing a too short acquisition time window (for example described in Ruthishauser et al. 2016). In this case, indeed, the instrument would underperform which would result in a negative bias.

**Lines 118-120:** "*For most temperate regions (10/13), a modern ≤ 100-MHz radar sounder could plausibly sound the maximum ice thicknesses of 95% of their glaciers (< 500 m; Fig. 2). For Alaska, Iceland and the Southern Andes, a lower-frequency (≤ 30 MHz) radar sounder remains necessary,...*"
This is the only section in the article, in which you give a hint how the empirical relationship you established could be used in practice. I would very much like to see a concluding remark at the end of the article, where you explain how you interpret your empirical relationship.
I don't agree that it is **plausible** that sounders with up to 100 MHz could sound the maximum thickness of this high amount of 95% of the glaciers in a region with glaciers of no more than 500 m ice thickness. I would rather say it is the maximum we can expect under perfect (and therefore unrealistic) circumstances.
Within the critical frequency range >10 MHz, the envelope is supported by only a few data points (mainly those with GlaThiDa ID's 507, 508, 2040, 2149, Fig. A). According to the discussions on Lines 110-113, the authors interpret the instrument performance for such points to be at their limit, whereas for other points, instruments either underperformed or where "unusually challenged by temperate ice". According to my experience in glacier thickness surveying, this challenge could occur on numerous glaciers. Therefore, it is not realistic that with a 100 MHz instrument the maximum ice thickness of 95% of the glaciers with ice thicknesses ≤ *500 m can be detected.*

If I would be in the position to design an instrument for surveying a large region with glaciers ≤ 500 m thick, I would, based on Figure 1a, choose a frequency of certainly no more than 50 MHz.

**Line 122:** Some of the histograms are difficult to read because lines are overprinted trough other histograms and because some colours are hard to differentiate. I suggest to display them separately with vertical offset and maybe in 2 or 3 columns.

**Line 157-159:** "*Multiple (≥ 3) antenna elements in a plane perpendicular to the platform's direction of travel are essential to resolving cross-track ambiguity in the direction of arrival of coherently recorded reflections,...*":
Additionally, it can be beneficial to use orthogonal pairs of antennas when the target glaciers are valley glaciers, as we have presented in our study by Langhammer et al. (2019).

[Figure]

*Figure A: Identical with Figure 1a by MacGregor et al. Black numbers indicate GlaThiDa ID's of points referred to in the reviewing comments.*

**References:**

Langhammer et al. (2019): "Glacier bed surveying with helicopter-borne dual-polarization ground-penetrating radar." Journal of Glaciology 65.249, 123-135.

Rutishauser et al. (2016). "Helicopter-borne ground-penetrating radar investigations on temperate alpine glaciers: A comparison of different systems and their abilities for bedrock mappingHelicopter GPR on temperate glaciers." Geophysics 81.1, WA119-WA129.

Welty, Ethan, et al. (2020): "Worldwide version-controlled database of glacier thickness observations." Earth System Science Data 12.4, 3039-3055.

---

## Referee Comment (RC2)

**Review: Brief communication: An empirical relation between center frequency and measured thickness for radar sounding of temperate glaciers (MacGregor et al.)**

**Topic:** The study analyzes previous records of radar-sounding measurements over temperate glaciers in order to investigate the relationship between the radar sounding frequency and the maximum measured ice thickness. Maximum ice thickness values are derived from the GlaThiDa inventory and compared to the center radar frequency used in the reported measurements. An empirical relationship is derived from this comparison, showing a decreasing maximum ice thickness with increasing radar frequency at a rate of ~500m per frequency decade. The study then calculates the maximum modeled ice thickness for all temperate glaciers in the RGI inventory, and uses the empirical envelope to predict the maximum radar frequency that could be used to survey different regions.

**General comments:** The manuscript is very well written, and the results are presented concise and clearly. In my opinion, the study makes a significant contribution to the radioglaciology community as it systematically analyzes, -and confirms, the previously assumed relationship between frequency and ice thickness over temperate glaciers. Furthermore, I think that this study provides useful insights when planning future radar sounding surveys. Overall, I think that the study is well-suited for a 'brief communication' in The Cryosphere, and only have a few recommendations/suggestions.

**Minor comments:**

**L35-36:** The following sentence structure is somewhat difficult to understand, maybe change to "… keep the signal-to-noise ratio  between the ice-bed reflection (signal) and  **cavity induced** volume scattering  (noise) **high.**"

**L95-96:** "… radar-sounding surveys have historically not always known beforehand where ice thickness is predicted to be greatest, …" The sampling bias resulting from this statement is not clear, I suggest clarifying this sentence.

**L98-99:** "Some glaciers we assume are temperate -… - are not.' This sentence reads as the authors know which glaciers were incorrectly assumed as temperate. I suggest changing the last part of the sentence to ' **maybe** not."

**Figure 2:** The figure is quite busy, and it is difficult to identify the ice thickness distribution for some survey regions. If possible, I suggest splitting the figure into a few separate panels, each including a few regions.

---

## Author Response (AR1)

**Response to reviews of "Brief Communication: An empirical relation between center frequency and measured thickness for radar sounding of temperate glaciers"**

Joseph A. MacGregor et al.
3 May 2021

We thank the editor and both reviewers for their thorough and constructive comments on this manuscript, especially given that these reviews were conducted amid the ongoing pandemic and associated workplace challenges. Below is our response to the reviewers' comments on the original manuscript submitted to The Cryosphere Discussions in January 2021. We have edited the manuscript in response and made a few other minor cosmetic changes, all with changes tracked. We've added an additional co-author, John Paden, who has provided further insight into suitable frequencies for radar sounders based on his experience.

Separately, after the discussion period closed, we received valuable external feedback from Laurent Mingo (Blue Systems Integration) on the potential ambiguity in our interpretation of the definition of center frequency. Briefly, he noted that the wavelength transmitted into the surrounding medium by a given physical antenna will be the same regardless of whether that medium is air or ice. However, the *frequency* of the resulting radio wave will differ depending on the medium's radio-wave velocity. Because the radio-wave velocity in ice is slower than that of air, the effective frequency of a given physical antenna is lower for a ground-based survey than an airborne one. In general, we expect that the reported frequency is for air, which is the case for airborne systems, commercial GPRs and likely to be the case for some (if not clearly all) custom-build ground-based systems. However, which frequency is reported in the literature is not always clear, so we have opted to clarify our assumption for ground-based surveys. This clarification does not materially alter our conclusions, but we have adjusted several statements to reflect this improved understanding.

Material comments are in blue and italicized, and our response in black follows each comment. For reviewer #1, some comments are grouped/reorganized to avoid repetition.

**Response to reviewer #1 (M. Grab)**

> *The relation between the central frequency of impulse radars and the penetration depth of the radar waves into glacier ice is complex, because it depends on various factors given by the nature of the glaciers such as water content in the ice, roughness of the glacier bed, or size and spatial density of crevasses, and on instrument parameters like the height of instrument employment above the ice or the transmitter power. Due to this complexity, no simple formula is known up to date for evaluating the central frequency required to investigate a glacier with a certain ice thickness.*

> *In recent years, a database has been compiled by Welty et al. (2020), containing an unprecedented number of ice thickness data from all over the world obtained from radar campaigns. In their study, MacGregor et al. made use of these data and established a simple log-linear relationship, that indicates the maximum ice thicknesses that can be surveyed with a given frequency. According to the authors and also to my knowledge, this is the first time, such an investigation has been done based on such a comprehensive amount of data.*

We thank the reviewer for their very positive comments. We are particularly glad that their perspective on the purpose of the manuscript is well aligned with what we hoped to achieve, i.e., not the last word on

relating radar center frequency and ice thickness, but a useful step in connecting the two by relating a lot of recently aggregated information.

*General Comment:*

*The study is well written and concise, therefore well-suited for a 'brief communication' in The Cryosphere journal. I agree with the authors (line 109) that a more complex relationship is not justified under consideration of the potential biases listed on lines 93-103.*

*I recommend the authors add a few sentences at the end of the article, where they summarize how they interpret their empirical relationship between sounding depth and frequency and how they would like to see it be used in practice (See also my minor comment about lines 118-120.*

*Lines 118-120: "For most temperate regions (10/13), a modern ≤ 100-MHz radar sounder could plausibly sound the maximum ice thicknesses of 95% of their glaciers (< 500 m; Fig. 2). For Alaska, Iceland and the Southern Andes, a lower-frequency (≤ 30 MHz) radar sounder remains necessary,..."*

*This is the only section in the article, in which you give a hint how the empirical relationship you established could be used in practice. I would very much like to see a concluding remark at the end of the article, where you explain how you interpret your empirical relationship.*

*I don't agree that it is plausible that sounders with up to 100 MHz could sound the maximum thickness of this high amount of 95% of the glaciers in a region with glaciers of no more than 500 m ice thickness. I would rather say it is the maximum we can expect under perfect (and therefore unrealistic) circumstances.*

*Within the critical frequency range >10 MHz, the envelope is supported by only a few data points (mainly those with GlaThiDa ID's 507, 508, 2040, 2149, Fig. A). According to the discussions on Lines 110-113, the authors interpret the instrument performance for such points to be at their limit, whereas for other points, instruments either underperformed or where "unusually challenged by temperate ice". According to my experience in glacier thickness surveying, this challenge could occur on numerous glaciers. Therefore, it is not realistic that with a 100 MHz instrument the maximum ice thickness of 95% of the glaciers with ice thicknesses ≤ 500 m can be detected.*

*If I would be in the position to design an instrument for surveying a large region with glaciers ≤ 500 m thick, I would, based on Figure 1a, choose a frequency of certainly no more than 50 MHz.*

We agree with the reviewer that the originally submitted manuscript was insufficiently clear in how this envelope could be interpreted. We've now added a new paragraph to the Discussion and Conclusions that clarifies our recommendations in terms of frequency given different survey scenarios. We've further reorganized some of the statements in this section to better distinguish the nature of this envelope and how we think it ought to be applied. We've replaced the term "plausible" with "possible", given the reviewer's dislike for it and because the former's definition can have negative connotations. Within the Figure 2 caption and in the text, we now clarify that this is meant to indicate a survey under ideal conditions.

*Minor Comments:*
*Line 18: "Newer airborne radar sounders generally outperform older, ground-based ones at comparable frequencies":*

*This finding cannot be deduced from the results presented in this manuscript. See comments about lines 87-88 below.*

*Lines 87-88: "Especially at higher frequencies (≥ 20 MHz), newer radar sounders (2000–onward) outperform older ones, which favored lower frequencies (≤ 10 MHz).":*

*This cannot be deduced from the data shown in Figure 1a. There are only two data points acquired before 2000 at frequencies ≥ 20 MHz: the two with GlaThiDa id's 334 and 2100 (see Fig A). The data point with id 334 is showing an ice thickness which is larger than all the thicknesses from newer campaigns at the same frequency and for temperate ice. Also in comparison with newer measurements at other frequencies the measured ice thickness of point 334 seems to be larger than average. The other data point with id 2100 shows a lower thickness of only around 120 meters but is even closer to the envelope. Either not all the data is shown (are there some symbols hidden by others in Fig 1a?) that leads to the conclusion that newer campaigns outperform older ones or this conclusion is wrong.*

All the available data is shown. The reviewer is correct that these statements concerning newer systems outperforming older ones is not always accurate, but we maintain that it is generally true within a broad frequency range (25–100 MHz), which we now specify. We included qualifiers to that statement previously (e.g., "tend to outperform"), but admit that these statements were insufficiently nuanced and did not capture the evolution well, leading to the confusion that the reviewer identified. We've reformulated both the abstract and this paragraph to better represent the observed patterns as follows.

Abstract sentence in question now reads:
"Between 25–100 MHz, newer airborne radar sounders generally outperform older, ground-based ones, so radar-sounder success is also influenced by system design and processing methods."

Paragraph in question now reads:
"Most radar-sounding surveys of temperate glaciers before 2000 (55 of 63) deployed systems with frequencies below 20 MHz, while about half (138 of 258) of modern surveys (2000–onward) used higher frequencies (≥ 20 MHz). At very low frequencies (< 3 MHz) and at higher ones (> 20 MHz), it is these modern radar sounders that sound unusually large ice thicknesses compared to older surveys at nearby frequencies. However, between 3–20 MHz ground-based surveys are predominant and sound the thickest ice. Between 25–100 MHz, newer airborne systems tend to outperform ground-based systems at comparable frequencies, and helicopter-borne systems tend to outperform fixed-wing systems (e.g., 100 MHz), presumably due to the former's slower platform speed and potentially lower altitude above ground level (Fig. 1b). Higher frequencies (≥ 60 MHz) can sound much thicker polythermal or polar ice than has been achieved for temperate glaciers, but this relative performance advantage for colder ice is reduced significantly below ~30 MHz, which may be due to lower antenna gains and increased environmental interference."

*Line 82: Caption of Figure 1-a: Only after studying Figure 1-b it becomes clear that grey symbols are data from cold glaciers. I suggest to add a comment in the caption.*

We've slightly deepened the color of these symbols to emphasize that they are blue and have added mention of their nature to both the legend to address this issue. We've also reworded the caption to flow better and capture this nuance.

*Lines 94-96: "2. While the situation is changing with the advent of globally modeled glacier thicknesses (e.g., Farinotti et al., 2019), radar-sounding surveys have historically not always known beforehand where ice thickness is predicted to be greatest, nor its expected value"*

*The reader can only guess how this leads to the "likely negative bias" to which you refer to on line 99. Please provide further explanations. Do you mean that ice thickness models provide information for planning radar surveys? If the glacier exhibits larger ice thicknesses than expected and researchers accidently use a radar system with too low penetration depth, their instruments perform at the limit and we would not expect a negative bias. Alternatively, researchers might acquire data while choosing a too short acquisition time window (for example described in Ruthishauser et al. 2016). In this case, indeed, the instrument would underperform which would result in a negative bias.*

We agree with the reviewer's assessment that this bias was not so clear, so we've reworded this bias. Our focus was on simply missing the maximum thickness if there's no good prediction of where it is beforehand, but the reviewer also identified other system selection and survey design considerations that would also induce this effect.

We've reworded this sentence to:
"Most radar-sounding surveys have rarely known beforehand where ice thickness is predicted to be greatest or its expected value there, so for a given glacier this location may not have been surveyed or a suitable radar sounder may not have been selected. (this situation has ameliorated recently with the advent of globally modeled glacier thicknesses, e.g., Farinotti et al., 2019)".

*Line 122: Some of the histograms are difficult to read because lines are overprinted trough other histograms and because some colours are hard to differentiate. I suggest to display them separately with vertical offset and maybe in 2 or 3 columns.*

Agreed. We've changed Figure 2 to be two panels now, with one panel representing the first six presumed temperate regions in sequential order of RGI region number and the second showing the remaining seven.

*Line 157-159: "Multiple (≥ 3) antenna elements in a plane perpendicular to the platform's direction of travel are essential to resolving cross-track ambiguity in the direction of arrival of coherently recorded reflections,...": Additionally, it can be beneficial to use orthogonal pairs of antennas when the target glaciers are valley glaciers, as we have presented in our study by Langhammer et al. (2019).*

Agreed, and thanks for drawing our attention to this highly relevant study, of which we were previously unaware. We've now added mention of this study and its orthogonal antenna strategy in the final paragraph: "Langhammer et al. (2019) recently demonstrated that two orthogonal pairs of helicopter-towed 25-MHz antennas can better sound temperate glaciers because the pseudo-scalar sum of the antenna response is less sensitive to unfavorable bed geometry."

**Response to reviewer #2 (anonymous)**

*General comments: The manuscript is very well written, and the results are presented concise and clearly. In my opinion, the study makes a significant contribution to the radioglaciology community as it systematically analyzes, -and confirms, the previously assumed relationship between frequency and ice thickness over temperate glaciers. Furthermore, I think that this study provides*

*useful insights when planning future radar sounding surveys. Overall, I think that the study is well-suited for a 'brief communication' in The Cryosphere, and only have a few recommendations/suggestions.*

We thank the reviewer for their clear and constructive review. Their summary demonstrates that what they gleaned from our study is exactly what we'd hoped to convey.

*Minor comments:*

*L35-36: The following sentence structure is somewhat difficult to understand, maybe change to "... keep the signal-to-noise ratio high between the ice-bed reflection (signal) and the cavity induced volume scattering arising from the cavities (noise) high."*

We agree and have simplified the last part of this sentence to: "between the ice–bed reflection (signal) and any cavity-induced volume scattering (noise)."

*L95-96: "... radar-sounding surveys have historically not always known beforehand where ice thickness is predicted to be greatest, ..." The sampling bias resulting from this statement is not clear, I suggest clarifying this sentence.*

Reviewer #1 had a similar comment on this particular bias. To clarify the nature of this sampling bias, we've changed this sentence to: "Most radar-sounding surveys have rarely known beforehand where ice thickness is predicted to be greatest or its expected value there, so for a given glacier this location may not have been surveyed or a suitable radar sounder may not have been selected. (this situation has ameliorated recently with the advent of globally modeled glacier thicknesses, e.g., Farinotti et al., 2019)"

*L98-99: "Some glaciers we assume are temperate -... - are not.' This sentence reads as the authors know which glaciers were incorrectly assumed as temperate. I suggest changing the last part of the sentence to 'are maybe not."*

Agreed and adjusted the final part of this sentence to "may not be temperate".

*Figure 2: The figure is quite busy, and it is difficult to identify the ice thickness distribution for some survey regions. If possible, I suggest splitting the figure into a few separate panels, each including a few regions.*

Reviewer #1 had a similar comment, so we've now broken out Figure 2 into two panels as per both reviewers' suggestions.

---

## Author Response (AR2)

**Response to reviews of "Brief Communication: An empirical relation between center frequency and measured thickness for radar sounding of temperate glaciers"**

Joseph A. MacGregor et al. 12 May 2021

We thank the editor for their additional comments on this manuscript, which have improved the manuscript's readability and value, and which we've addressed below in the same format as before.

The only content-related point that I have links back to the first general comment of Reviewer #1: as was the case for the reviewer, I wonder whether you want to use some more words of caution when phrasing the main conclusion, i.e. when reporting the central frequency by which the majority of the glaciers of a particular region can be sounded. If I read Fig. 2a correctly, for Central Europe this conclusion suggests that a system of 200HMz or higher would be sufficient. I'm not suggesting this conclusion to be wrong, but if I had to invest money in a ice thickness sounding campaign, I would probably venture in the field with a system that has a frequency lower than that. If you have a similar gut feeling, please consider amending the text in a few instances.

We've taken this comment to heart and adjusted several sentences to clarify that the upper envelope represents an ideal scenario. More importantly, we have adjusted both figures to include a lower envelope to better capture uncertainty in our present understanding and also some of our concluding statements.

The remaining comments are of purely technical nature. A set of specific remarks is given at the end of this document, whilst two general points are related to the format requirements of a Brief Communication:

1) With around 170 words, the abstract is 70% longer than it should be for the particular format (cf. the entry "Brief Communication" at <a href="https://www.the=cryosphere.net/about/manuscript\_types.html">https://www.the=cryosphere.net/about/manuscript\_types.html</a>). Although I imagine that there will be some tolerance when typesetting, please see if you can save some words here and there.

We've substantially shortened the abstract to 125 words, which we believe meets the spirit of the Brief communication.

2) A similar comment applies for the overall length of the document. Whilst I don't think there is a need for removing content, a careful check might allow for avoiding some particularly wordy construct (e.g. L. 27: "[...] temperate (i.e., at or near the pressure-melting point throughout)" could simply read "[...] temperate (i.e., at the pressure-melting point)".

We agree and have further reviewed the manuscript with this concern in mind, deleting numerous words, phrases and sentences. While we were unable to ultimately reduce page length, we cut down the wordiness and flourishes while adding context with the lower envelope mentioned above.

L. 18: Please check and possibly clarify the wording "~500m per frequency decade". I first thought that "frequency" might just need to be removed but I see that the exact same wording is used at L. 125.

Based on our review of the applied terminology, this term ("frequency decade") appears appropriate.

*L.* 79-81: Please clarify that the stated "440m" (and other values) refer to the maximal thickness (I caught myself wondering if that might be the total profile length, or something else).

Now clarified as "maximum thicknesses".

L. 99: Consider rewording "55 of 63" into something like "53 surveys out of 63".

Changed to "(55 of 63 surveys)"

L. 109: Please clarify what "This synthesis" is referring to. Possibly simply reword into "The above synthesis".

Changed to "The above synthesis".

L. 111-112: Please check the last part of the sentence: I'm not sure what "there" is referring to.

Simply removed "there" (the majority of us thought it was fine, but one co-author also disliked it in an earlier draft).

L. 112-113: Please remove the full stop before the parenthesis.

Done.

L. 121: Consider starting a new sentence for the part beginning with "but".

Split up sentence but did so earlier.

L. 131-134: Please check whether you can simplify this sentence. Possibly split the sentence into separate ones.

We've made a minor adjustment to this sentence to address this concern but decided to keep the sentence mostly as is.

L. 139: Please check the use of "which": it might be interpreted as referring to the subject of the sentence, i.e. to "this comparison", causing the question about how this can be said to be "larger than" something.

Changed ", which" to "and" to address this concern.

L. 154-155: Please consider a slight reword: to me, the maximum thickness is "only reached at a single point" by definition, whilst the sentence seems to suggest some surprise for this assertion.

Removed the qualifier "generally".

Figure 1: (1) Caption: please add "(blueish symbols)" (or similar) at the end of the second sentence, i.e. the one starting with "Cold glaciers are either those from...". (2) Can the "blue arrows" in panel

"(b)" be enlarged? It took me a while to spot them, and I first wondered whether the wording "represented by blue arrows" might be a leftover from previous versions.

Clarified nature of blue symbols in caption and darkened/enlarged blue arrows.

Figure 2: Please use the caption to clarify that the two panels are used to differentiate between two sets of RGI regions, and that this split is only for reasons of readability.

Done.

**Brief communication: An empirical relation between center frequency and measured thickness for radar sounding of temperate glaciers**

Joseph A. MacGregor1, Michael Studinger1, Emily Arnold2,3, Carlton J. Leuschen3, Fernando Rodríguez-5 Morales3, John D. Paden3

1Cryospheric Sciences Laboratory (Code 615), NASA Goddard Space Flight Center, Greenbelt, Maryland, 20771, United States of America

2Aerospace Engineering Dept., The University of Kansas, Lawrence, Kansas, 66045, United States of America

3Center for Remote Sensing of Ice Sheets, The University of Kansas, Lawrence, Kansas, 66045, United States of America

**10**

Correspondence to: Joseph A. MacGregor (joseph.a.macgregor@nasa.gov)

Abstract. Radar sounding of the thickness of temperate glaciers is challenged by substantial volume scattering, surface scattering and high attenuation rates. Lower frequency radar sounders are often deployed to mitigate these effects, but the lack of a global synthesis of their success, limits progress in system and survey design. Here we extend a recent global compilation

15 of glacier thickness measurements (GlaThiDa) with the center frequency for radar-sounding surveys. From a maximum reported thickness of ~1500 m near 1 MHz, the maximum thickness sounded decreases by ~500 m per frequency decade. Between 25–100 MHz, newer airborne radar sounders generally outperform older, ground-based ones, Based on globally modeled glacier thicknesses, we conclude that a multi-element, < 30-MHz airborne radar sounder could survey most temperate glaciers more efficiently.

**20 1 Introduction**

Measuring the thickness of Earth's mountain glaciers is essential for advancing understanding of their volume, flow and future amid ongoing anthropogenic warming, consequent mass loss and contribution to sea-level rise (e.g., Farinotti et al., 2019; Zemp et al., 2019). Radar sounding is unambiguously the preferred method for most surveys of glacier thickness (Welty et al., 2020). However, most mountain glaciers outside the polar regions are either observed or assumed to be temperate (i.e., at the pressure-melting point), and radar sounding of such ice is more challenging than for polar ice sheets (where most of ice column is well below the pressure-melting point) or polythermal glaciers (e.g., Pritchard et al., 2020). Three main factors conspire to cause this challenge: 1. Englacial water in pore spaces or fractures, increasing volume scattering (e.g., Fountain et al., 2005); 2. More common crevassing and supraglacial debris, increasing surface scattering (e.g., Herreid and Pellicciotti, 2020); and 3.

| Deleted: more                                                                                                                    |
|----------------------------------------------------------------------------------------------------------------------------------|
| Deleted: ing than for polar ice sheets,                                                                                          |
| Deleted: due to the former's greater                                                                                             |
| Deleted: (englacial water)                                                                                                       |
| Deleted: (crevasses and debris)                                                                                                  |
| Deleted: dielectric                                                                                                              |
| Deleted: (warmer ice)                                                                                                            |
| Deleted: (~1–100 MHz)                                                                                                     |
| Deleted: commonly                                                                                                                |
| Deleted: ,                                                                                                                       |
| Deleted: of existing radar-sounding surveys of temperate glaciers                                                         |
| Deleted: use                                                                                                                     |
| Deleted: recent                                                                                                                  |
| Deleted: synthesis                                                                                                               |
| Deleted: to evaluate the relation between the radar center frequency and maximum thickness                                |
| Deleted: sounded                                                                                                                 |
| Deleted: with increasing frequency                                                                                               |
| Deleted: , so radar-sounder success is also influenced by system design and processing methods                            |
| Deleted: with a center frequency of $\leq 30$ MHz                                                                         |
| Deleted: more efficiently than presently available systems                                                                       |
| Deleted: , due to its logistical efficiency relative to other methods while achieving satisfactory precision and accuracy |
| Deleted: or near                                                                                                                 |
| Deleted: throughout                                                                                                              |
| Deleted: substantially                                                                                                           |

**1**

Warmer ice, increasing the englacial dielectric attenuation rate (e.g., Stillman et al., 2013).

Watts and England (1976) described what may be the primary challenge in radar sounding of temperate ice: meter-scale, water-filled englacial cavities that efficiently scatter incident radio waves where the ratio of those cavities' radius to the radar's englacial wavelength exceeds ~0.1. Their analysis favored center frequencies  $\leq \sim 10$  MHz to increase the signal-to-clutter ratio

- 60
- between the ice-bed reflection (signal) and any cavity-induced volume scattering (clutter). Their lucid description of this challenge motivated the development of numerous Jow-frequency, radar sounders (e.g., Watts and Wright, 1981; Fountain and Jacobel, 1997; Conway et al., 2009; Mingo and Flowers, 2010; Rignot et al., 2013; Arnold et al., 2018; Björnsson and Pálsson, 2020). However, subsequent advances in available hardware, system design and processing demonstrated that higher-frequency (> 10 MHz) radar sounders can also sound hundreds of meters of temperate ice (e.g., Rutishauser et al., 2016;
- 65 Langhammer et al., 2019; Pritchard et al., 2020). No synthesis yet exists of the success of these radar sounders as a function of center frequency, limiting our ability to identify outstanding opportunities in system and survey design for more efficient sounding of temperate glaciers. Here we evaluate past and potential radar-sounder performance by examining recent global compilations of both observed and modeled glacier thickness.

**2 Data and methods**

70 We primarily use three data sources in this study: 1. The maximum reported ice thickness for individual surveys compiled in the Glacier Thickness Database (GlaThiDa) version 3.1.0 (Welty et al., 2020); 2. The consensus modeled thickness estimates for all glaciers on Earth (Farinotti et al., 2019); 3. The first-order regions, glacier locations, areas and identification numbers of the Randolph Glacier Inventory (RGI) version 6 (RGI Consortium, 2017).

We extend GlaThiDa with one additional field: the center frequency of the deployed radar sounder for surveys that used this method (Supplementary Information). While englacial wavelength is likely the more fundamental physical property affecting radar-sounder success (Watts and England, 1976), it is the center frequency that is most often reported and / straightforward to compile. In most cases, we could determine the value of this field directly from other metadata compiled by / GlaThiDa. In dozens of cases, we reviewed the primary source for the survey to determine the center frequency, assuming that the deployed radar sounder with the lowest center frequency was that which detected the maximum ice thickness reported, if /

80 that was not stated explicitly. In 210 cases (4% of radar-sounding surveys in GlaThiDa), we were unable to determine the center frequency.

For ground-based surveys, a potential source of ambiguity is whether the reported center frequency is for the antenna, radiating in air or through ice, which can be challenging to determine from available metadata. The wavelengths radiated by a given antenna are determined by its physical size, design and input signal; these wavelengths are the same if that antenna is on

85 the ground or airborne, but their frequencies will differ because the propagation velocity through air and ice differs. If the reported center frequency is for a ground-based system is that for transmission through ice, then it will be lower by a factor of ~1.79 (the radio-frequency index of refraction for ice) than that of most other reported frequencies (e.g., Mingo and Flowers, 2010), which are typically that through air. Here we assume that all reported center frequencies are that through air.

[revised manuscript text omitted]

170 abundant and hinders radar performance; 5. Some reported center frequencies for ground-based surveys may be for ice, rather than for air as we have assumed (Sect. 2); and 6. Some glaciers we assume are temperate – due to their regional setting and the lack of contraindicating observations – may not be temperate. The first five biases are likely negative, i.e., they induce an

| Deleted: of the same dataset   |  |
|--------------------------------|--|
| Deleted: only                  |  |
| Deleted: (temperate or "cold") |  |
| Deleted: "                     |  |
| Deleted: "                     |  |

| De | leted: | comparable   |
|----|--------|--------------|
|    |        | comparatione |

| Dele | ted: is              |
|------|----------------------|
| Dele | ted: are potentially |
| Dele | ted: s               |
| Dele | ted: relative to     |
| Dele | ted: radar-sounding  |
| Dele | ted: there           |
| Dele | ted: is              |
| Dele | ted: .               |

underestimate of the maximum ice thickness that could be sounded at a given frequency, while the sixth bias is positive with the opposite effect. Separately, off-nadir surface clutter is a well-known source of ambiguity in identification of the ice-bed

190 reflection within valley glaciers (e.g., Holt et al., 2006). Most surveys had no direct method for surface-clutter discrimination (e.g., Conway et al., 2009; Rignot et al., 2013), but its bias can be positive or negative depending on glacier geometry and survey design.

Based on the above synthesis, we identify a simple envelope for the maximum temperate ice thickness sounded across the more than two decades of frequency spanned by deployed radar sounders (Fig. 1b). For the frequency range shown (1-1000 MHz), this empirical relation is  $H_{max} = H_{max}^0 - 500 \log_{10} f$ , where  $H_{max}$  is the maximum ice thickness in meters.  $H_{max}^0$  is the maximum ice thickness in meters at 1 MHz and f is the center frequency in megahertz. This linear relation between ice thickness and logarithmic frequency was selected because it captures the predominant trend but does not overfit thickness

maxima with as-of-yet unjustified complexity, While precise, uncertainty bounds for this envelope cannot yet be specified, we present approximate lower and upper bounds for this envelope using  $H_{max}^0$  values of 1250 and 1500 m, respectively, although we note that two surveys report sounding temperate ice slightly thicker than even the upper envelope suggests. The envelope's value lies in its indirect synthesis of the previously mentioned factors that challenge sounding of temperate ice, its identification

of radar-sounder frequencies that may be performing near a natural or present technical limit (e.g.,  $\sim 2-3$ ,  $\sim 25-30$  and 100 MHz), and others where radar sounders have either underperformed or may be unusually challenged by temperate ice ( $\sim 5-20$  MHz).

- Using this empirical envelope, we can then crudely relate the maximum modeled thickness distribution for each predominantly temperate RGI region to a suitable radar-sounder frequency (Fig. 2). We consider only larger glaciers (RGI-reported area ≥ 5 km2) that are more likely to be targeted for radar-sounding surveys. This comparison highlights the maximum possible frequency that could sound most glaciers in these regions under ideal conditions (upper bound of envelope) and the more conservative suggested frequency (lower bound), both of which are larger than recommended previously (Watts and
- 210 England, 1976). For most temperate regions (10 of 13), this analysis suggests that a modern ≤ 100-MHz radar sounder could potentially sound the maximum thickness of 95% of their glaciers (< 500 m; Fig. 2), although a ≤ 30-MHz system is a more conservative suggestion. For thicker glaciers in Alaska, Iceland and the Southern Andes, a lower-frequency (≤ 30 MHz) radar sounder remains necessary for many glaciers and ≤ 10 MHz may be more suitable, as observed in practice (e.g., Conway et al., 2009; Björnsson and Pálsson, 2020).

5

215

| Deleted: is                                                                                                                                                                                |
|--------------------------------------------------------------------------------------------------------------------------------------------------------------------------------------------|
| Deleted: of                                                                                                                                                                                |
| Deleted: for temperate glaciers                                                                                                                                                            |
| Deleted: range                                                                                                                                                                             |
| Deleted: Within                                                                                                                                                                            |
| Deleted: is                                                                                                                                                                                |
| Deleted: ,                                                                                                                                                                                 |
| Deleted: the maximum possible ice thickness that can be sounded decreases at a rate of ~500 m per frequency decade, descending from ~1500 m at 1 MHz. For the frequency range shown |
| Deleted: equivalent to                                                                                                                                                                     |
| Deleted: 1500                                                                                                                                                                              |
| Deleted: empirical envelope has several limitations: 1. It is simply a                                                                                                              |
| Deleted: , which                                                                                                                                                                           |
| Deleted: and                                                                                                                                                                               |
| Deleted: ;                                                                                                                                                                                 |
| Deleted: 2. N                                                                                                                                                                              |
| Deleted: o meaningful                                                                                                                                                                      |
| Deleted: presently                                                                                                                                                                         |
| Deleted: ; and 3. T                                                                                                                                                                        |
| Deleted: is                                                                                                                                                                                |
| Deleted: for radar sounders                                                                                                                                                                |
| Deleted: glacierized                                                                                                                                                                       |
| Deleted: maximum possible                                                                                                                                                                  |
| Deleted: in each region                                                                                                                                                                    |
| Deleted: range of frequencies                                                                                                                                                              |
| Deleted: possibly                                                                                                                                                                          |
| Deleted: , which is                                                                                                                                                                        |
| Deleted: assumed                                                                                                                                                                           |
| Deleted: ice                                                                                                                                                                               |
| Deleted: se                                                                                                                                                                                |
| Deleted: most                                                                                                                                                                              |
| Deleted: is                                                                                                                                                                                |
| Deleted: ; Fig. 2                                                                                                                                                                          |

Figure 2: Distribution (lines) and 95th percentile (circles) of maximum modeled ice thickness for all larger (≥ 5 km2) glaciers in RGI regions assumed to contain mostly temperate glaciers. RGI regions are separated into two panels for readability. Bin interval is 25 m. Upper horizontal axes are equivalent to the lower and upper bounds of the empirical envelope in Fig. 1, e.g., for a region whose 95th percentile maximum ice thickness is ~500 m, the maximum center frequency of a suitable radar sounder for that region is ~100 MHz, while ~30 MHz is a more conservative suggested frequency.

We note two contrasting caveats to this analysis: 1. While the modeled maximum thicknesses do not appear to be biased significantly relative to measured maxima (+62 ± 197 m for the 36% of GlaThiDa surveys we could confidently match using glacier names to a modeled glacier in the RGI inventory), it is not uncommon for radar-sounding surveys of ice masses to report ice thicknesses greater than predicted beforehand, so any survey design based on Fig. 2 should assume a negative model bias and favor the lower envelope; and 2. The maximum modeled thickness is only reached at a single point on a glacier, so surveys aiming to measure glacier volume must also consider the ability to resolve smaller thicknesses at a satisfactory resolution. This trade-off could favor a higher center frequency, for which a larger bandwidth is easier to achieve, potentially

**4 Discussion and conclusions**

resulting in a finer range resolution.

270

Our analysis of GlaThiDa-compiled thicknesses challenges the conventional wisdom that only low-frequency ( $\leq 10$  MHz) radar sounders are suitable for sounding temperate glaciers that are hundreds of meters thick (Fig. 1). An empirical envelope derived from this synthesis suggests that – for most temperate glaciers on Earth – higher frequencies are indeed appropriate for radar sounding thereof, assuming suitable system design (Fig. 2). For example, this envelope suggests a higher upper limit (~30 MHz) on potentially suitable radar sounders for sounding temperate ice up to ~750 m thick (Fig. 1), but a more conservative interpretation still suggests that ~500 m is possible at this frequency. Advances in hardware and system design could further increase this range, e.g., solid-state transmit/receive switches, higher peak-transmit powers and platform-aware

| Deleted                          | evaluation                                                                          |
|----------------------------------|-------------------------------------------------------------------------------------|
| Deleted                          | evaluation                                                                          |
|                                  |                                                                                     |
| Deleted                          | - 77                                                                                |
| Deleted                          | : I`                                                                                |
|                                  |                                                                                     |
| Deleted                          | 0                                                                                   |
| Deleted
Deleted               | : 0
: and could be suitable for sounding the large majority of                   |
| Deleted
Deleted
Earth's gl | : 0
: and could be suitable for sounding the large majority o
aciers (Fig. 2) |

numerical optimization of antenna configurations (Arnold et al., 2020). While a full accounting of the physical underpinnings of the empirical envelope we identified was beyond the scope of this study, the envelope cannot be explained by the relatively weak dispersion of the radio-frequency dielectric attenuation rate (MacGregor et al., 2015), so volume and surface scattering 290 are the more likely controls.

295

This empirical envelope can help further balance a radar-sounding survey's scientific objectives with the system to be deployed. For example, selecting a system toward the lower end of frequency range considered (e.g.,  $\leq 10$  MHz) is only necessary if the objective is to sound the thickest temperate ice in a handful of regions. Alternatively, a higher frequency (> 20 MHz) radar sounder could be deployed to more finely resolve glacier volume at the possible expense of not sounding the thickest portions of some glaciers. Not all glaciers are created equal, and in some cases an exceptionally crevassed surface, thicker supraglacial debris or abundant englacial water will continue to necessitate the use of lower frequencies to ensure successful sounding.

- Models of glacier thickness increasingly incorporate mass conservation and global satellite remote-sensing datasets, but there remains an outstanding need for additional thickness measurements to both validate those models and refine their 300 underlying assumptions (e.g., Farinotti et al., 2019; Pelto et al., 2020). Even the most spatially extensive ground-based radarsounding survey of a glacierized region can be easily dwarfed by a single airborne survey - assuming that survey's radar sounder can match the performance of the ground-based system (e.g., Pritchard et al., 2020). Langhammer et al. (2019) demonstrated recently that two orthogonal pairs of helicopter-towed 25-MHz antennas can better sound temperate glaciers because the sum of the antenna response is less sensitive to unfavorable bed geometry. Alternatively, a slightly higher
- 305 frequency (e.g., 30 MHz) translates to a dipole antenna that is  $\leq 5$  m long, a sufficiently small dimension that several such elements could be mounted on a fixed-wing aircraft (e.g., Arnold et al., 2018). More than two such antenna elements in a plane perpendicular to the platform's direction of travel are essential to resolving cross-track ambiguity in the direction of arrival of coherently recorded reflections, i.e., "swath mapping" (e.g., Holschuh et al., 2020). By transforming some of what is presently discarded as noise (discontinuous near-bed subsurface reflections not attributable to surface clutter) into useful signal

310 (geolocated, off-nadir ice-bed reflections), swath mapping could substantially expand our ability to measure glacier thickness efficiently.

Author contribution. JAM initiated this study, led the analysis and drafted the manuscript, MS aided in the study design, analysis and manuscript preparation. EA, CJL, FRM and JDP aided in the analysis and manuscript preparation.

Data and code availability. The Supplementary Information associated with this article contains the new metadata generated 315 by this study in a format similar to GlaThiDa v3.1.0. The analysis was performed using MATLAB R2021a and both its Deleted: simple

| Deleted: We suggest using t                |
|--------------------------------------------|
| Deleted: as a guide to                     |
| Deleted: radio-                            |
| Deleted: 5                                 |
| Deleted: 2                                 |
| Deleted: a larger                          |
| Deleted: or rough                          |
| Deleted: or simply exceptionally thick ice |
| Deleted: are                               |
| Deleted: ing                               |
| Deleted: those                             |
| Deleted: models'                           |
| Deleted: The spatial coverage of e         |

| Deleted: pseudo-scalar              |  |
|-------------------------------------|--|
| Deleted: is sufficiently small that |  |
| Deleted: potentially                |  |

Mapping and Statistics & Machine Learning toolboxes. The script used to perform the analysis and generate the figures is available at: <a href="https://github.com/joemacgregor/misc/blob/main/temperate\_sounding.m">https://github.com/joemacgregor/misc/blob/main/temperate\_sounding.m</a>.

340 Competing interests. JAM is a member of the editorial board of this journal.

Acknowledgements. We thank the NASA/GSFC Strategic Science program for supporting this study. We thank the authors of and contributors to GlaThiDa, Farinotti et al. (2019) and the RGI Consortium for making our study possible through open distribution of well-documented, homogenized datasets. We thank R. Hale, J.W. Holt, H. Jiskoot, H. Machguth, L. Mingo and M. Truffer for valuable discussions. We thank scientific editor D. Farinotti, M. Grab and an anonymous reviewer for constructive reviews that improved this manuscript.

**References**

[revised manuscript text omitted]